# Opiate Withdrawal-Associated Esotropia: A Case Report and Systematic Review

Varun Kasula [1,*], Brody M. Fogleman [2], Maaya Dev [3], Tyler Rizzieri [2], Corinne O'Brien [1] and Rupa Shetty [4]

1   Department of Psychiatry, Campbell University School of Osteopathic Medicine, Lillington, NC 27546, USA
2   Department of Psychiatry, Edward Via College of Osteopathic Medicine—Carolinas, Spartanburg, SC 29303, USA
3   Department of Psychiatry, Kansas City University College of Osteopathic Medicine, Kansas City 64106, MO, USA
4   Lighthouse Behavioral Health Hospital, Conway, SC 29526, USA
*   Correspondence: v_kasula0902@email.campbell.edu

**Abstract:** Esotropia, which is the medial deviation of one or both eyes, is a rare withdrawal symptom that has been associated with opiate addiction. We report a case of a 36-year-old female patient who developed acute-onset esotropia and diplopia after self-admission to a psychiatric facility for fentanyl addiction treatment and a systematic review of this rare presentation. A search of four databases (PubMed, Scopus, Embase, and Google Scholar) was conducted as of January 2024. We found 15 documented cases of opiate withdrawal-associated esotropia, with an average age of 27.2 years and an average time between last use and symptom onset of 5.61 days. The most common symptom was diplopia, especially binocular diplopia, and the majority of cases resolved without pharmacologic intervention. Considering the current opioid crisis, our systematic review and case report add valuable insight into the less-explored neurological and ophthalmological consequences of opiate withdrawal, a condition that should always be considered in cases of acute or chronic onset esotropia.

**Keywords:** opiates; heroin; fentanyl; withdrawal symptoms; esotropia; strabismus; diplopia

## 1. Introduction

Opioid abuse is a global concern, contributing to 60 million people struggling with its addictive effects and more than 100,000 deaths every year [1]. Specifically in the United States, opioids were involved in 80,411 overdose deaths in 2021 [2]. Some estimates of the total economic burden of opiate use disorder on the US healthcare system are as high as USD 95.43B, which amounts to 7.86% of all hospital expenditures [3]. Drug addiction is a chronic disease, commonly associated with multiple bouts of relapse, in which drug administration becomes the stimulus for one's behavior regardless of the adverse effects. As tolerance increases, so does reward-seeking behavior. This can lead to physical dependence on the drug, which is characterized by physiological adaptations within the body due to prolonged, repetitive exposure to a substance that eventually cause the body to require it for normal function; discontinuation of the substance will precipitate withdrawal symptoms. In opioid use disorder, common withdrawal symptoms include myalgia, nausea, vomiting, diarrhea, tachycardia, restlessness, diaphoresis, and ocular symptoms such as diplopia, ptosis, and lacrimation [4].

Although ocular manifestations are common in those experiencing opioid withdrawal, strabismus characterized by acute esotropia is seldom reported in the literature [5]. Strabismus is a misalignment of one or both eyes. Esotropia is one type of strabismus in which one or both eyes deviate medially, giving an individual a "cross-eyed" appearance. This effect was first discovered in soldiers returning from the Vietnam War; however, due to limited studies, the pathophysiology detailing the underlying mechanism of how opioid

withdrawal causes this ocular symptom is not well understood [6,7] This case describes a patient with acute esotropia, hypothesized to be due to fentanyl withdrawal. Additionally, to the best of our knowledge, this is the first systematic review and narrative analysis on this topic, aiming to enhance overall awareness and understanding of this rare opiate withdrawal-associated symptom.

## 2. Case Report

A 36-year-old female with a past medical history of opioid abuse, major depressive disorder, generalized anxiety disorder, bipolar depression, post-traumatic stress disorder, abdominal abscess, and chronic back pain was admitted to the psychiatric facility for opioid use disorder. She reported a history of heroin use since adolescence. Her last use was four months prior. She also reported using fentanyl daily for several years, and her last use was two days prior to admission to the inpatient psychiatric unit. The patient also reported daily alprazolam use for the past year, with the most recent use three days prior. On admission, she reported auditory and visual hallucinations, sweating, nausea, vomiting, poor impulse control, irritability, decreased energy, and loss of interest in activities. Vital signs included blood pressure 136/80 mmHg, temperature 37.0 °C (98.6 °F), heart rate 98 beats/minute, and respiratory rate 16 breaths/minute.

She was voluntarily admitted to the facility for detoxification management. Initial physical examination revealed a sluggish, disheveled, and lethargic patient who was oriented to person and situation only. Cardiac and pulmonary examinations were unremarkable. Urine drug screen (UDS) returned positive for benzodiazepines only. Complete blood count (CBC) and complete metabolic panel (CMP) were unremarkable. The patient also reported that she was homeless, spending some time sleeping outdoors, and did not have a stable support system. She reported no previous hospitalizations or surgeries.

The patient was started on the following medications to manage cravings and withdrawal symptoms: transdermal nicotine patch, clonidine, trazodone, and lorazepam. Buprenorphine/naloxone was initiated on hospital day four.

She responded well to clinical management and had a remarkable improvement in her symptoms until hospital day five when she began to experience blurred vision and dizziness. Vitals at that time revealed blood pressure 115/70 mmHg, temperature 36.6 °C (97.9 °F), heart rate 90 beats/minute, respiratory rate 17 breaths/minute, and oxygen saturation 98% on ambient air. Physical examination revealed left-eye esotropia, sluggish pupillary response of the left eye, no facial asymmetry, normal muscle strength in all four extremities, and a known abdominal abscess in the left lower quadrant. Her right eye appeared normal with no esotropia at the time with an appropriate pupillary response. She reported no tenderness to palpation of the frontal or maxillary sinuses, and denied ocular pain, headache, loss of sensation, weakness, loss of extremity proprioception, chest discomfort, or difficulty breathing.

Due to the acuity and abrupt onset of ocular manifestations, the patient was transferred to the nearest emergency department for further evaluation. Computerized tomography (CT) of the head without contrast did not reveal any acute intracranial abnormalities. CBC and CMP were again unremarkable. The patient was then transferred back to the psychiatric facility for continuation of the opioid detoxification protocol. The patient was provided with an eye patch, which she said helped her blurry vision.

On hospital day six, the patient continued to have blurred vision when not using the eye patch and developed new-onset esotropia in the right eye. We did not observe the esotropia occurring simultaneously; the patient's esotropia was observed to affect either the left or the right eye independently when the eye patch was not being used. Neurological examination remained benign.

She was declared medically stable and discharged from the facility on hospital day seven. She received a short-term prescription of buprenorphine/naloxone 2 mg/0.5 mg twice daily to reduce cravings and other withdrawal symptoms, as well as trazodone 100 mg for nightly use to manage insomnia. She was advised to follow up with her

outpatient psychiatrist for continued management of her opioid use disorder and consult an ophthalmologist for further evaluation of suspected cranial nerve (CN) VI palsy.

We attempted to contact the patient on multiple occasions following discharge to follow up regarding her diplopia and esotropia but were unable to make contact.

### 3. Systematic Review

This systematic review was conducted per the Preferred Reporting Items for Systematic Reviews and Meta-analyses (PRISMA) 2020 guidelines (Figure 1 and Supplementary Figure S1) [8]. The protocol of this review was recorded in the INPLASY register under the number INPLASY202430129.

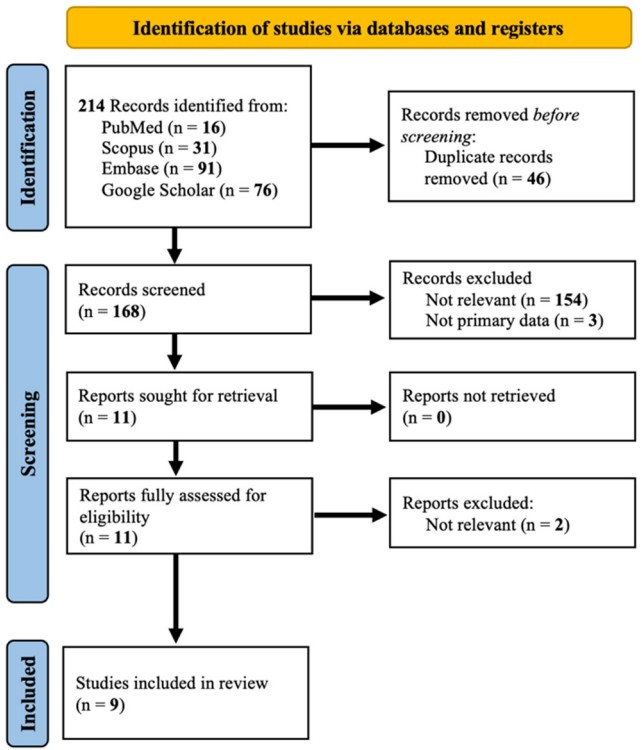

**Figure 1.** PRISMA diagram for study selection process.

### 3.1. Materials and Methods

#### 3.1.1. Search Criteria and Screening Strategy

A database search was conducted in January 2024 in PubMed, Scopus, Embase, and Google Scholar using combinations of the following terms: "opiates, heroin, fentanyl, withdrawal, withdrawal symptoms, esotropia, strabismus, exotropia, diplopia, nystagmus" (Supplementary Table S1). No other filters were used in the systematic search. Results were imported into Rayyan, a free online software used for systematic reviews [9]. Articles were screened by two independent and blinded reviewers (CO and MD) and conflicts were resolved by a third independent reviewer (VK). Articles were included based on the following criteria: (1) must be a full-text paper that was published in a peer-reviewed journal; (2) studies that were published in English; and (3) studies that investigated esotropia, strabismus, or similar oculomotor symptoms related to opiate withdrawal.

#### 3.1.2. Data Extraction and Variables of Interest

The included articles were subjected to a data extraction process using Microsoft Excel. Variables regarding study characteristics such as journal, year, and type of drug use were collected. Further, information on length of drug use, last use before onset of symptoms, and ocular findings were also recorded. Finally, information was collected regarding ocular treatments, non-ocular treatments, outcomes, and most recent follow-up.

### 3.1.3. Quality Assessment

The Joanna Briggs Institute (JBI) Critical Appraisal Checklist for Case Reports was used to conduct a rigorous quality assessment of the included studies after the screening process [10]. The checklist consists of eight different items and an overall judgement on the strength of the study (Figure 2). The quality assessment was independently completed by MD and TR, and all conflicts were resolved by a third independent reviewer, CO.

| Authors | Were the patient's demographic characteristics clearly described? | Was the patient's history clearly described and presented as a timeline? | Was the current clinical condition of the patient on presentation clearly described? | Were diagnostic tests or assessment methods and the results clearly described? | Was the intervention(s) or treatment procedure(s) clearly described? | Was the post-intervention clinical condition clearly described? | Were adverse events (harms) or unanticipated events identified and described? | Does the case report provide takeaway lessons? | Overall |
|---|---|---|---|---|---|---|---|---|---|
| Wu et al. (2008) | Yes | Yes | Yes | Yes | Yes | Yes | Yes | Yes | 8 Yes |
| Satish et al. (2018) | Yes | Yes | Yes | Yes | No | No | Unsure | Yes | 5 Yes, 2 No, 1 Unsure |
| Hakimi et al. (2016) | Yes | Yes | Yes | Yes | Yes | Yes | Unsure | Yes | 7 Yes, 1 Unsure |
| Czyz et al. (2015) | Yes | Yes | Yes | Yes | Yes | Yes | Yes | Yes | 8 Yes |
| Shiferaw et al. (2015) | Yes | Yes | Yes | Yes | Yes | Yes | Unsure | Yes | 7 Yes, 1 Unsure |
| Rabin (2015) | Yes | Yes | Yes | Yes | Yes | Yes | Yes | Yes | 8 Yes |
| Mattoo et al. (2012) | Yes | Yes | Yes | Yes | Yes | Yes | Unsure | Yes | 7 Yes, 1 Unsure |
| Sutter et al. (2003) | Yes | No | No | Yes | No | Yes | Unsure | Yes | 4 Yes, 3 No, 1 Unsure |
| Firth (2001) | Yes | Yes | Yes | Yes | Yes | No | Unsure | Yes | 6 Yes, 1 No, 1 Unsure |

**Figure 2.** Quality assessment of included studies using the JBI Critical Appraisal Checklist for Case Reports [7,11–18].

### 3.2. Results

Due to the relatively rare incidence of this complication, a systematic review was also performed to understand the context of our patient's condition relative to what has been reported in the literature. There were 214 results initially obtained using the search string, with 46 duplicate articles that were resolved. Of the 168 remaining articles, 157 were excluded following dual independent abstract review due to irrelevance or lack of primary data. Full-text review of the remaining 11 papers after abstract screening yielded nine primary articles comprising 15 patients (Figure 1).

The average age of the included patients was 27.2 years, with nine males and six females (Table 1). The most common drug intoxication reported was heroin (14 of 15 cases), with 1 case reporting diplopia due to withdrawal from dextropropoxyphene. The average length of drug use was three years, and six patients had a history of relapse. Data regarding the length of time between last reported drug use before the onset of symptoms were calculated with a mean length of 4.69 days (Range: 2–8) and a median of 3 days. Regarding symptoms, four patients reported binocular diplopia while ten patients reported other variants of esotropia. Although imaging was performed in most reports (seven papers), there were no acute findings in any of the cases. The most common treatment modality employed for ocular symptoms was pilocarpine and Fresnel prisms (20% each). Also, no treatment was administered in four cases (26.7%), and the treatment protocol was not reported at all in five cases (33.3%). The resolution of symptoms was documented in 10 of 15 cases.

**Table 1.** Demographic information, drug use and medical history, and outcomes of the patients in the included studies (M: male; F: female; NR: not reported).

| Study | n | Sex | Age | Drug Used | Length of Drug Use (Years) | History of Relapse (y/n) | Last Use before Onset of Symptoms (Days) | Ocular Findings and Symptoms | Other Symptoms | Treatment for Ocular Symptoms | Outcome at last Follow-Up | Most Recent Follow-Up (Months) |
|---|---|---|---|---|---|---|---|---|---|---|---|---|
| Wu et al. (2008) [11] | 1 | M | 31 | Heroin, hydrocodone | 2 | Yes | 7 | Binocular horizontal diplopia | NR | Occlusion patch on one eye, Pilocarpine 1% | Persistent diplopia, only mild improvement in symptoms | 1.23 |
| Satish et al. (2018) [12] | 1 | M | 34 | Heroin, marijuana | NR | Yes | 8 | Right-eye esotropia, diplopia | None | None | Resolution of diplopia | NR |
| Hakimi et al. (2016) [13] | 1 | F | 25 | Heroin | 1 | No | 2 | Binocular horizontal diplopia | None | None | Constant distance diplopia unchanged | 1 |
| Czyz et al. (2015) [14] | 1 | M | 31 | Heroin | NR | Yes | 3 | Binocular diplopia | None | Pilocarpine 2% | Pilocarpine was discontinued after 2 weeks due to lack of success. Diplopia self-resolved 5 weeks later. | 1.63 |
| Shiferaw et al. (2015) [7] | 1 | F | 22 | Heroin | NR | No | 7 | Diplopia, left-eye esotropia, fine horizontal nystagmus | NR | None | Resolution of diplopia and esotropia | 2.33 |
| Rabin (2015) [15] | 1 | F | 22 | Heroin, marijuana, alprazolam | 1 | Yes | 8 | Diplopia, blurry vision, esotropia, nystagmus | Nausea, vomiting, diarrhea, weight loss, headaches | Pilocarpine 2% used once but caused blurry vision | Resolution of esotropia | 6.4 |
| Mattoo et al. (2012) [16] | 1 | M | 30 | Dextropropoxyphene | 8 | Yes | 3 | Binocular horizontal diplopia, blurry vision | Rhinorrhoea, lacrimation, aches and pain, loose stools | Fogged glasses | Resolution of diplopia | 0.23 |
| Sutter et al. (2003) [17] | 1 | M | 22 | Heroin | NR | NR | NR | Small-angle right esotropia in right and downgaze | NR | NR | Unchanged | 1 |
| | 1 | F | 23 | Heroin | NR | Yes | 3 | Large-angle right comitant esotropia for distance and near | NR | NR | Resolution of esotropia | 5 |
| | 1 | F | 32 | Heroin | NR | NR | 3 | Right comitant esotropia with varying angles | NR | NR | Resolution of esotropia | 0.5 |
| | 1 | M | 25 | Heroin | NR | NR | 2 | Left comitant esotropia | NR | NR | Resolution of esotropia after restarting heroin use | 1 |
| | 1 | M | 31 | Heroin, cocaine | NR | NR | 4 | Left comitant esotropia with distance | NR | NR | Improved esotropia | 0.4 |
| Firth (2001) [18] | 1 | M | 27 | Heroin | NR | NR | NR | Left to alternating esotropia with diplopia | NR | Fresnel prism | Intermittent diplopia | 7 |
| | 1 | M | 31 | Heroin | NR | NR | 8 | Intermittent horizontal diplopia | NR | Fresnel prism | NR | NR |
| | 1 | F | 22 | Heroin | NR | NR | 3 | Moderate alternating esotropia | NR | Fresnel prism | Improved esotropia | 1.5 |

## 4. Discussion

The patient under consideration experienced a sudden onset of esotropic strabismus during detoxification treatment for opioid misuse. Initial differential diagnosis included cerebrovascular accident (CVA), mass-related central nervous system (CNS) dysfunction, idiopathic intracranial hypertension (i.e., pseudotumor cerebri), Lyme disease, and opiate withdrawal-induced CN VI palsy presenting as left-sided strabismus. However, as the clinical picture evolved, the absence of evidence pointing to a structural CNS abnormality prompted a focused exploration into the potential association with opiate use or withdrawal. Our literature searches exploring acute-onset esotropia identified opioid withdrawal as a possible explanation for our patient's presentation, although supporting evidence for this is mostly limited to case reports, as indicated by the systematic search.

Given our patient's history of substance abuse and onset of ocular motor symptoms following her admission, there was suspicion that the withdrawal phase from fentanyl or introduction of buprenorphine/naloxone therapy may be the cause of her ocular symptoms. Although UDS was negative for opioids, it is not surprising given that standard screening tests do not reliably detect synthetic opiates like fentanyl [19]. On further investigation, the patient did mention prior episodes of blurry vision while using fentanyl; however, she denied ever noticing misalignment of her eyes. Our initial uncertainty of the etiology underlined our decision to transfer the patient to the nearest emergency department for further evaluation to rule out other more serious etiologies of ocular dysfunction. We also considered the possibility of Lyme disease, as it has been known to be the cause of a variety of unique neurologic clinical manifestations, including strabismus [20]. Since the patient was homeless with recent activity in the woods and had a concurrent abdominal abscess, we considered the possibility of a tick bite that could have resulted in the subsequent abdominal abscess formation alongside unique CNS manifestations. The lack of other clinical manifestations consistent with this diagnosis led us to not pursue further diagnostic investigations regarding this possibility.

At the time of ocular dysfunction, our patient had not used fentanyl within the past seven days and had started buprenorphine/naloxone therapy the same day. Therefore, it is unclear whether the introduction of buprenorphine/naloxone therapy, which is a partial opioid agonist and antagonist combination, caused the symptomology or if the withdrawal phase from higher doses of opioids is the more likely cause. Currently, there is no widely accepted association between buprenorphine/naloxone therapy and oculomotor dysfunction, making this an unlikely explanation of this patient's strabismus.

Including the case reported here, our systematic review identified 16 total cases of opiate withdrawal-associated esotropic strabismus. To the best of our knowledge, our patient represents the first documented case of fentanyl withdrawal-associated esotropia. Among the 15 cases currently documented in the literature, all but one were attributed to heroin, with the only exception being linked to dextropropoxyphene. It is notable that our patient had a history of intermittent heroin usage dating back to her adolescence. As fentanyl misuse continues to increase, it is important to be aware of these types of ocular symptoms in fentanyl users as well [21]. It is also worth noting that our patient stopped using alprazolam at the same time as fentanyl, but there does not seem to be any literature suggesting that benzodiazepine withdrawal can cause acute onset esotropia. Unfortunately, our patient was lost to follow-up, so it is unknown whether any further treatments were initiated or if her symptoms resolved. However, the results of our systematic review suggest that the self-resolution of symptoms is the most common course of action with opiate withdrawal-associated oculomotor dysfunction. In fact, for all three cases in which pilocarpine (1% or 2%) was used as a treatment, it either did not completely ameliorate symptoms or potentiated blurry vision. In fact, any resolution of symptoms was only noted well after discontinuation of the medication (Table 1). Thus, although we acknowledge more data are required, it appears that pilocarpine may not be universally effective in treating oculomotor dysfunction caused by opiate withdrawal.

Currently, the limited nature of clinical information and investigations relating to the treatment of oculomotor dysfunction in the setting of opiate withdrawal restricts the ability to provide concretely supported treatment recommendations. In the present case, our patient was able to tolerate the symptomatology without major implications on her quality of life. Since we were unable to establish contact with the patient following her discharge from our facility, it is unclear if her ocular motor disturbances have completely resolved or if she had received any further management. The treatment modalities utilized in the majority of the included studies in this systematic review appeared to employ standard conservative management or simply did not report a treatment intervention at all, making it appear that most individuals can tolerate the symptomology until spontaneous resolution. Some of the cases did utilize eye patches or the pharmacologic alternative, pilocarpine, to force the dysfunctional ocular muscles to restore ocular competence. Eye patches, fogged glasses, and pilocarpine are thought to work via a similar physiological mechanism. Simply put, eye patches and fogged glasses remove or decrease the visual acuity of the good eye and force the contralateral, incompetent eye to establish focus on an object instead of relying on the opposing eye to do the work. Pilocarpine, a cycloplegic agent, is theorized to act as a non-physical patch by inducing blurriness in the good eye, which subsequently influences the incompetent eye to regain motor competence to focus on an object [22]. However, the effect size of these differences between any of these utilized treatment modalities, if any at all, remains unclear at this time due to the limited ability to perform statistical significance testing on such a low number of data points. To further refine the rates at which pharmacologic intervention is needed in these cases, as well as the efficacy of these treatment modalities, more robust large-scale clinical studies are recommended.

Interestingly, the average age of males exhibiting this symptom was 29.1 years, while the average age of females was 24.3 years. Although the sample size is not large enough to determine if this age difference is statistically significant, one possible explanation for why women may experience this withdrawal symptom at a younger age is known as the "telescoping effect". This refers to a phenomenon observed in substance use disorder in which females progress more rapidly from initial use to development of dependence to admission [23,24]. One recent study has suggested that females may experience withdrawal-like symptoms faster than males as well, adding another potential dimension to the telescoping effect; however, more research needs to be conducted on this topic, especially in the context of opiate use disorder [25]. Although the telescoping effect may play a role in the observed age difference between sexes regarding withdrawal symptoms, more research is necessary to draw a definitive conclusion, especially in the context of opiate use disorder. This is further challenged by the fact that our patient is both female and somewhat older compared to the current cases in the literature regarding this topic. However, her history of chronic heroin use may have built up compensatory mechanisms that delayed the precipitation of this symptom.

There are multiple proposed mechanisms for opiate withdrawal-induced esotropia. One mechanism is the decompensation of a pre-existing hypermetropic condition that had been previously compensated [11]. Hypermetropia, or farsightedness, often necessitates increased accommodative effort to maintain clear vision and binocular alignment. In the context of opiate use, the induced miosis narrows the diameter of the pupil, thus creating a smaller aperture and allowing more of the visual field to be in focus with less accommodative effort [14,26]. Thus, opiate use can augment the compensatory accommodation in hypermetropic individuals and can even improve their near-sighted vision [5]. Upon withdrawal, this compensatory state is disrupted, leading to an acute manifestation of esotropia as the demand for accommodative convergence suddenly increases.

Another proposed mechanism is that the abrupt cessation of opiate intake can precipitate a parasympatholytic or hyperadrenergic state characterized by pupillary dilation and ciliary muscle paralysis [17]. This can lead to a failure of the physiological processes that sustain binocular motor fusion mechanisms and manifest as esotropic deviation [14]. In other reported cases, the physiologic stress of withdrawal, akin to a "physical or psychic

shock", has been thought to destabilize the neuromuscular control of eye movements, reinforcing the strabismus [11,27].

In addition to these functional disturbances, structural predispositions may also contribute to the development of esotropia in the context of opiate withdrawal. Neuroanatomically, the supraoculomotor area of the midbrain, pretectum, and rostral superior colliculus constitute a neural hub integral to the regulation of near-response and vergence movements [28–30]. Opiate use may alter the normal activity of these pre-motor vergence neurons, leading to an imbalance in the convergence and divergence equilibrium—an imbalance that may be exacerbated during withdrawal—and thus resulting in the observed esotropia [18]. Moreover, a recent study found that abstinent opiate users demonstrate reduced midbrain resting-state functional connectivity, an effect that was most pronounced in individuals who recently quit [31]. These findings may underscore a state of neuroplastic reorganization or dysfunction in the neuronal circuits of the midbrain following the cessation of opiate use. Such alterations in midbrain connectivity could undermine the stability of the vergence system, predisposing individuals to the decompensation of oculomotor control mechanisms, and culminating in the manifestation of esotropia.

Furthermore, similar to how opiate withdrawal can cause esotropia, opiate use may also cause exotropia, or the outward deviation of the eyes. To the best of our knowledge, there are four documented cases of exotropia in active opiate users; however, the sparsity of the literature on this topic is likely due to difficulty in gaining access to patients at the time of illegal drug use [13,17,32]. Notably, the patient in a study conducted by Hakimi et al. reported experiencing horizontal binocular diplopia for 3 months, which the authors suspect was due to heroin-induced exotropic deviation, prior to presentation for acute onset esotropia 2 days following last heroin use [13]. A proposed mechanism by which this may occur is that the ability of opiate use to decrease the drive for accommodative convergence may lead to a relative overcompensation through which an exotropic deviation can manifest. Thus, if opiate use can cause exotropia, it is plausible that opiate withdrawal has the potential to reverse this symptom and result in esotropia [7,11].

*Limitations*

There are several limitations of this case report that warrant consideration. First, the inherent nature of case reports constrains the generalizability of the findings as they typically describe unique or rare instances. Additionally, the patient was lost to follow-up upon her discharge, thus impeding the ability to provide outcome data on this topic. Pertaining to the systematic review, the small sample size of accessible documented cases is a limiting factor as it may not capture the full spectrum of clinical presentations or the prevalence of acute-onset esotropia in the context of opiate withdrawal. It is notable that the two included studies that yielded multiple cases were retrospective case series studies [17,18]. Thus, we recommend conducting additional case series and even retrospective cohort studies, as they may be especially fruitful in terms of their contribution to the overall data available on this topic. Furthermore, relying exclusively on published case reports may introduce publication bias, as cases with more typical presentations or less severe symptoms may be under-reported. Lastly, as all the included studies are case reports, there is an inherent lack of standardized diagnostic criteria and treatment protocols, which could lead to variability in the reporting and management of the condition. The heterogeneity in data-reporting standards among the included studies presents a challenge for synthesizing a consistent overview of the condition, further increasing the necessity for more published literature on this topic to provide definitive conclusions and recommendations.

**5. Conclusions**

Our systematic review and accompanying case report have underscored esotropia as a rare but clinically significant presentation of opiate withdrawal. This phenomenon poses a diagnostic challenge and calls for a high degree of clinical vigilance, particularly in patients with a history of opiate use presenting with visual disturbances. The pathophysiology

underlying this condition appears to be multifaceted, involving neuroadaptive changes within the central nervous system and possibly exacerbated by pre-existing ocular or systemic conditions. The identification of esotropia in the context of opiate withdrawal in our review highlights the importance of comprehensive post-withdrawal monitoring and understanding of potential oculomotor dysfunction in such patients.

In light of the current opioid crisis, our systematic review and case report add valuable insight into the less-explored neurological and ophthalmological consequences of opiate withdrawal. Future investigations with larger cohorts are needed to better understand the incidence, management strategies, and long-term outcomes of opiate withdrawal-associated esotropia. Healthcare professionals should consider opiate withdrawal in differential diagnoses of acute onset esotropia, ensuring early recognition and appropriate management.

**Supplementary Materials:** The following supporting information can be downloaded at: https://www.mdpi.com/article/10.3390/psychiatryint5020016/s1, Figure S1: Search terms used for each database. Table S1: Search terms used for each database.

**Author Contributions:** Conceptualization, V.K., B.M.F., T.R. and R.S.; Methodology, V.K., M.D. and C.O.; Formal Analysis, V.K., B.M.F. and M.D., Investigation, V.K., B.M.F., M.D. and T.R.; Resources, V.K. and B.M.F.; Data Curation, B.M.F., M.D. and C.O., Writing—Original Draft Preparation, V.K., B.M.F., M.D., T.R. and C.O.; Writing—Review and Editing, V.K., B.M.F., M.D. and R.S.; Visualization, V.K., M.D. and T.R. All authors have read and agreed to the published version of the manuscript.

**Funding:** This research received no external funding.

**Institutional Review Board Statement:** Ethical review and approval were waived for this study due to the study being a single case report.

**Informed Consent Statement:** Informed consent was obtained from all subjects involved in the study.

**Data Availability Statement:** The data presented in this study are available on request from the corresponding author due to ethical reason.

**Conflicts of Interest:** The authors declare no relevant financial or non-financial interests to disclose. The authors also declare that no funds, grants, or other support were received during the preparation of this manuscript.

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
