# Peer review of "Opiate Withdrawal-Associated Esotropia: A Case Report and Systematic Review"

_2673-5318, doi:10.3390/psychiatryint5020016_

Round 1
Reviewer 1 Report
Comments and Suggestions for Authors
This paper describes a case study of opiate withdrawal-associated esotropia paired with a systematic review of other similar case reports.
The manuscript is generally clearly presented. The case study is carefully documented and the systematic review has been carried out according to the recommended PRISMA guidelines. The combination of the case study and the systematic review makes the manuscript a very useful contribution to the literature both for clinicians and researchers in this field.
There are a few points that could be addressed by the authors:
1) The authors use the term dependence however 'dependence' is no longer a recommended term according to DSM V. If the authors wish to use this term they would need to define its meaning within this manuscript.
2) The authors may wish to emphasise that the last two papers they include in the systematic review Sutter et (ref 17) and Firth (ref 18) are retrospective studies that appear to be done by searching records of their own institutions. Have the authors considered doing a retrospective study or would they recommend that more retrospective studies should be done?
3) In table 1 the authors describe 5 patients originally reported by Sutter et al (Reference 17) . In the their paper Sutter et al actually describe 7 patients, two of which have exotropia as opposed to esotropia . Towards the end of the discussion the authors briefly mention exotropia and indeed refer to the same paper of Sutter, as one the source for the only 4 known cases (it may be useful to explicitly mention this as it may not be noticed by the reader). As mentioned in point 2 could not this be considered further motivation for recommending retrospective studies to identify more cases of both exotropia and esotropia that simply have not been reported as case studies?
4) In table 1 , column 8 the authors report the last use of drug before onset of symptoms. For the majority of the cases it is 2-4 days but there are notable exceptions of some case with 10,11 and 14 days. The authors may wish to comment on this more. Could it be that these patients had a delayed onset of withdrawal rather than a delayed onset of esotropia?
Reviewer 2 Report
Comments and Suggestions for Authors
This study is based on one patient that showed symptoms of esotropia where either both or at least one pupil of the eye is misaligned which is a withdrawal symptom of opiate consumption. I do not have any major concerns about the study and the manuscript is written in a more or less decent manner. My only concern with this study is that the authors based their study off one patient. I am aware of the literature search the authors conducted that was based off similar symptoms observed in patients. I would still try to get more replicates i.e. more data from other patients and analyze them before trying to publish the data.
Reviewer 3 Report
Comments and Suggestions for Authors
Nice work enjoyed reading the manuscript. Appreciate the nice table. I have made just very few minor comments for your consideration.
Good luck.

Reviewer 4 Report
Comments and Suggestions for Authors
This manuscript nicely provided a case report and a literature review concerning acute onset esotropia in the context of opiate withdrawal, thus potentially calling for vigilance about this clinical presentation and motivating relevant mechanism studies, although it seems no active or proactive treatment is suggested. Only one minor formality issue is noticed, i.e., the inconsistency of capitalized or non-capitalized column titles in Table 1. Accordingly, I fully support its publication.
Round 2
Reviewer 2 Report
Comments and Suggestions for Authors
In my previous review report, I had mentioned my reluctance to accept this manuscript for publication and my main rationale behind the same was lack of biological replicates. Although I am still not fully convinced that a scientific study, no matter how rare the topic of interest is, needs to have enough background information and biological replicates. Only then it is considered to be good science. But having gone through the latest manuscript and also the justification provided by the authors against my initial argument about lack of data, I would endorse this manuscript for publication.
Author Response
We greatly appreciate the reviewer's response and understand the perspective put forth by the reviewer. Based on the response, we have no further revisions to add to the paper and extend our gratitude to the reviewer for taking the time to review and comment on our manuscript.